# Insights into the Composition of a Co-Culture of 10 Probiotic Strains (OMNi BiOTiC^®^ AAD10) and Effects of Its Postbiotic Culture Supernatant

**DOI:** 10.3390/nu14061194

**Published:** 2022-03-11

**Authors:** Bernhard Kienesberger, Beate Obermüller, Georg Singer, Christoph Arneitz, Paolo Gasparella, Ingeborg Klymiuk, Angela Horvath, Vanessa Stadlbauer, Christoph Magnes, Elmar Zügner, Pablo López-García, Slave Trajanoski, Wolfram Miekisch, Patricia Fuchs, Holger Till, Christoph Castellani

**Affiliations:** 1Department of Paediatric and Adolescent Surgery, Medical University of Graz, 8034 Graz, Austria; bernhard.kienesberger@medunigraz.at (B.K.); georg.singer@medunigraz.at (G.S.); c.arneitz@medunigraz.at (C.A.); paolo.gasparella@medunigraz.at (P.G.); holger.till@medunigraz.at (H.T.); christoph.castellani@medunigraz.at (C.C.); 2Department of Cell Biology, Histology and Embryology, Medical University of Graz, 8034 Graz, Austria; ingeborg.klymiuk@medunigraz.at; 3Division of Gastroenterology and Hepatology, Department of Internal Medicine, Medical University of Graz, 8034 Graz, Austria; angela.horvath@medunigraz.at (A.H.); vanessa.stadlbauer@medunigraz.at (V.S.); 4Center of Biomarker Research (CBmed), 8034 Graz, Austria; pablo.lopez-garcia@cbmed.at; 5Health—Institute for Biomedicine and Health Sciences, Joanneum Research, 8010 Graz, Austria; christoph.magnes@joanneum.at (C.M.); elmar.zuegner@joanneum.at (E.Z.); 6Core Facility Computational Bioanalytics, Medical University of Graz, 8034 Graz, Austria; slave.trajanoski@medunigraz.at; 7Department of Anesthesiology and Intensive Care, Rostock University Medical Center, 18057 Rostock, Germany; wolfram.miekisch@uni-rostock.de (W.M.); patricia.fuchs@uni-rostock.de (P.F.)

**Keywords:** postbiotic, microbiome, volatile organic compound, shotgun keyword, susceptibility, culture

## Abstract

Background: We aimed to gain insights in a co-culture of 10 bacteria and their postbiotic supernatant. Methods: Abundances and gene expression were monitored by shotgun analysis. The supernatant was characterized by liquid chromatography mass spectroscopy (LC-MS) and gas chromatography mass spectroscopy (GC-MS). Supernatant was harvested after 48 h (S48) and 196 h (S196). Susceptibility testing included nine bacteria and *C. albicans*. Bagg albino (BALBc) mice were fed with supernatant or culture medium. Fecal samples were obtained for 16S analysis. Results: A time-dependent decrease of the relative abundances and gene expression of *L. salivarius*, *L. paracasei*, *E. faecium* and *B. longum/lactis* and an increase of *L. plantarum* were observed. Substances in LC-MS were predominantly allocated to groups amino acids/peptides/metabolites and nucleotides/metabolites, relating to gene expression. Fumaric, panthotenic, 9,3-methyl-2-oxovaleric, malic and aspartic acid, cytidine monophosphate, orotidine, phosphoserine, creatine, tryptophan correlated to culture time. Supernatant had no effect against anaerobic bacteria. S48 was reactive against *S. epidermidis*, *L. monocytogenes*, *P. aeruginosae*, *E. faecium* and *C. albicans*. S196 against *S. epidermidis* and *Str. agalactiae*. In vivo S48/S196 had no effect on alpha/beta diversity. Linear discriminant analysis effect size (LEfSe) and analysis of composition of microbiomes (ANCOM) revealed an increase of *Anaeroplasma* and *Faecalibacterium prausnitzii*. Conclusions: The postbiotic supernatant had positive antibacterial and antifungal effects in vitro and promoted the growth of distinct bacteria in vivo.

## 1. Introduction

The importance of the intestinal microbiome has gained wide scientific interest in health and disease, and improving human health through modulation of microbial interactions during all phases of life has become increasingly important [1]. Intestinal bacteria have, amongst others, been associated with the fermentation of short-chained fatty acids (SCFA) from non-digestible oligosaccharides, synthesis of secondary and tertiary from primary bile-acids, and modulation of the intestinal immune system [2].

The composition of the intestinal microbiome is susceptible to nutritional changes or medication. Chronic diseases such as tumor-associated cachexia, inflammatory bowel disease or type two diabetes have been associated with altered compositions of the intestinal microbiome [3]. Therefore, modification of the intestinal bacterial composition towards a “healthier” microbiome has become attractive as possible therapeutic or supportive therapy approach. Apart from dietary modifications and stool transplantation, this could be achieved by nutritional supplementation with pre-, pro- or synbiotics [4,5,6]. The probiotic effect of lactobacilli, for instance, has been tested in in vivo experiments yielding beneficial results in various diseases [7]. Currently, many food supplementations contain a variety of different probiotic strains.

OMNi BiOTiC^®^ AAD10 (distributed by Institut AllergoSan (Graz, Austria) and produced by Winclove Probiotics B.V. (Amsterdam, The Netherlands) a GMP facility for manufacturing dietary supplements complying with NSF/ANSI standard 173-2008 and certified according to ISO 22000:2005) is a commercially available probiotic food supplement composed of 10 different probiotic strains: *Lactobacillus* (L.) *acidophilus* W55 and W37, *L. paracasei* W72, *L. rhamnosus* W71, *L. salivarius* W24, *L. plantarum* W62, *Enterococcus faecium* W54, *Bifidobacterium (B.) bifidum* W23, *B. lactis* W18 and *B. longum* W51. All of these strains belong to species with reported beneficial effects (reviewed in [8]), for instance in preventing constipation, travelers’ diarrhea, antibiotic-associated diarrhea, prevention and treatment of necrotizing enterocolitis, reduction of radiation induced diarrhea, or reducing the risk of food allergies [9,10,11,12,13,14]. Compared to single strain probiotics, synergistic combinations of probiotics may be beneficial. In this regard, a previous study could clearly demonstrate advantages of the multi-species probiotic VSL#3—With a composition (*S. thermophilus*, *E. faecium*, *B. breve*, *B. infantis*, *B. longum*, *L. acidophilus*, *L. plantarum*, *L. casei* and *L. delbrueckii*) close to OMNi BiOTiC^®^ AAD10, compared to single and multi-strain probiotics in case of AAD [15]. The focus of OMNi BiOTiC^®^ AAD10 lies in the therapy and prevention of antibiotic-associated diarrhea (AAD). In detail, various clinical trials could prove its potential in therapy and prevention of AAD [16,17,18].

While generally considered as safe, there is a minimal risk for adverse side effects especially in vulnerable patient cohorts such as neonates, elderly, or immunocompromised patients [8]. In this regard probiotic bacteremia, fungemia, spread of antimicrobial resistance and altered long-term immune responses have been described [8,19,20,21,22,23]. In these patients, a helpful alternative for the treatment of intestinal dysbiosis is warranted [24]. Therefore, alternatives to live microorganisms such as bacteriocins, antimicrobial bacterial metabolites or postbiotics may be of interest to manage dysbiosis [25]. At present, however, it is unclear if the probiotic effect is caused by the bacteria themselves or by bacterial metabolites/components as for instance short chained fatty acids (SCFAs), secondary bile acids or microbe associated molecular patterns as lipopolysaccharide or peptidoglycan. This question fueled the research in the field of postbiotics.

Postbiotics are defined as preparation of inanimate microorganisms and/or their components that confers a health benefit on their host [1]. They include any substance released or produced through the metabolic activity of microorganism, which has a direct or indirect beneficial effect for the host [26,27]. These substances can be either cell free supernatants (CFS), exopolysaccharides, enzymes, cell wall fragments, short chained fatty acids (SCFAs) or bacterial lysates [27]. CFS contain electrolytes, carbohydrates, amino acids, lactate, bacterial toxins (as bacteriocins and antimicrobial peptides) and other bacterial metabolites. The major benefits of postbiotics are their inherent stability in industrial processes and storage, intellectual property protection (as no live microorganisms can be isolated from the postbiotic) and their better safety profile compared to probiotics [1].

In bacterial culture experiments, the growth of pathogenic germs such as *Enterobacter*, *Escherichia*, *Salmonella* and *Klebsiella* has been successfully inhibited by treatment with culture supernatants of single probiotic bacteria [28,29,30]. At present, there is very limited clinical data on postbiotics. However, it could be shown that oral administration of inactivated lactic acid bacteria was effective in the treatment of *Helicobacter pylori* infection, reduced symptoms in patients with irritable bowel disease or chronic unexplained diarrhea [31,32,33]. As such, the conglomerate of bacterial defensive substances and metabolic products of a known culture composition may selectively (as opposed to conventional antibiotics) influence the intestinal microbiome and therefore be used as targeted therapy against harmful microorganisms.

At present, however, there is only limited information about the behavior of combinations of different probiotic strains (as present in a variety of commercially available food supplements) in co-cultures. Therefore, the aim of the present study was to investigate a co-culture of 10 different probiotic strains contained in the probiotic OMNi BiOTiC^®^ AAD10 in culture and to gain insights into the composition and in vivo effects of the postbiotic culture supernatant.

## 2. Materials and Methods

### 2.1. Bacterial Culture and Supernatant Production

OMNi BiOTiC^®^ AAD10 was kindly provided by the Institut Allergosan (Graz, Austria). This probiotic contains *Bifidobacterium* (B.) lactis W18, B. longum W51, B. bifidum W23, Enterococcus faecium W54, *Lactobacillus* (L.) acidophilus W55 and W37, *L. paracasei* W72, *L. plantarum* W62, *L. rhamnosus* W71 and *L.*
*salivarius* W24. For the culture 1 g OMNi BiOTiC^®^ AAD10 (equalling 4.5 × 10^10^ CFU) was dissolved in 15 mL of sterile filtered SuperPure Water (Milli-Q Direct 8, Merck Millipore, Darmstadt, Germany) and stored at room temperature in darkness for 10 min of activation time. 500 mL sterile De Man, Rogosa and Sharpe (MRS) broth in a 1 l Duran glass bottle were flooded with N_2_ and inoculated with 8.3 mL of OMNi BiOTiC^®^ AAD10 solution. No other consumables were added in the further course of the culture. 400 µL sample were harvested at timepoint 0 (infection of the culture) and after 8, 12, 24, 36, 48, 96, 144, 168, and 196 h to monitor bacterial activity. OD600 measurement was conducted with a SpectraMax Plus 384 (Molecular devices, San Jose, CA, USA) as duplicates. Further pH measurement and colony forming units (CFU) counts were performed to monitor the growth of the culture. For CFU quantification, a dilution series in MRS broth was prepared until a 10^6^ dilution was reached and then seeded to MRS agar plates as duplicates. The plates were incubated under anaerobic conditions (O_2_ Absorber Thermo Fisher Scientific, Waldham, MA, USA) for 48 h at 37 °C at 120 rpm (Excella E24 Incubator Shaker Series, New Brunswick Scientific, Canada Scientific Lab Systems Inc., Guelph, ON, Canada). Then the number of colonies were counted and documented.

To gain insights into the behavior of the co-culture, 12 mL of culture were sampled at 8, 12, 24, 48, 96 and 196 h for shotgun, metabolomics, and volatile organic compound (VOC) analysis. For shotgun analysis, 1.5 mL of culture liquid were harvested to Eppendorf vials and stored at −80 °C. For metabolomics (2 times 2 mL, stored at −80 °C) and volatilomics (2 times 2 mL, immediately sent for analysis) the culture liquid was centrifuged and filtered to stop the bacterial reaction. Room air samples were obtained at each time point to rule out contamination in the VOC analysis.

For the in vivo application and susceptibility testing 170 mL of the supernatant were harvested after 48 h as production step 1, sterile filtrated, aliquoted, and stored at −80 °C. The remaining supernatant in the bottle was further cultivated as described above and harvested after 196 h as production step 2, sterile filtrated, aliquoted, and stored at −80 °C.

### 2.2. Susceptibility Testing

Bacteria for susceptibility testing were obtained from Aurosan GmbH, Essen, Germany. *Clostridium difficile* (ATCC 700057), *Listeria monocytogenes* (ATCC 15313), *Escherichia coli* (ATCC 25922), *Enterococcus faecium* (ATCC 27270), *Staphylococcus (S.) aureus* (ATCC 29213), *Staphylococcus (S.) epidermidis* (ATCC 12228), *Streptococcus (Str.) agalactiae* (ATCC 13813), *Pseudomonas (P.) aeruginosa* (ATCC 27853), and *Propionibacterium (Pr.) acnes* (ATCC 6919) were chosen for bacterial resistance testing. *Candida albicans* was kindly provided by A.H. Detailed information on bacterial cultures is given in the Appendix A.

Each microorganism was cultured on agar plates with suitable growth medium [34] (Appendix A). Using a stencil, 9 disks for resistance testing (BD Sensi-Disc™, Becton, Dickinson and Company, Franklin Lakes, NJ, USA) were placed on each culture plate using a prepared scheme. Each disk was either treated with 20 µL of pure supernatant, a 1:2 or 1:4 dilution of the supernatant, cooked supernatant, supernatant buffered to pH 7.0 (with 1 n NaOH), supernatant mixed with 1 n HCl (1:1) or supernatant treated with 1 mg/mL Proteinase K (Carl Roth, Germany) for 1 h prior to application. Culture medium of supernatant production (MRS) served as negative control and either vancomycin or piperacillin/tazobactam as positive control. Plates were then incubated at 37 °C for 24 h at room air in case of aerobic and for 48 h under anaerobic conditions (O_2_ Absorber Thermo Fisher Scientific, Waldham, MA, USA) in case of anaerobic bacteria. Thereafter, plates were photographed, and inhibition zones were determined with ImageJ 2.0.0-rc-69/1.52p (ImageJ opensource image processing software, http://imagej.net/Contributors, accessed on 7 May 2021). A detailed description is presented in the Appendix A.

### 2.3. VOC Headspace Analysis

All samples were immediately sent to the partner via overnight express for gas chromatography/mass spectroscopy. VOC analysis was performed in the headspace of samples as previously reported [35,36,37]. VOCs were pre-concentrated with a commercially available solid phase micro extraction (SPME) fiber (carboxen/polymethylsiloxane, Sulpeco, Bellefonte, PA, USA). An Agilent 7890 A gas chromatograph (GC) coupled to an Agilent 5975 C inert XL mass selective detector (MSD) was used to separate and identify the VOCs desorbed from the SPME device. Detected marker substances were identified from a mass spectral library (National Institute of Standards and Technology 2005; NIST 2005, Gatesburg, PA, USA) and by retention time matching. In case the median of the room air samples exceeded 30% of the median of the headspace samples a possible contamination was recorded and the substance was excluded from further analysis. The area responses of a selected m/q ratio at a defined retention time for each substance were recorded, integrated and used for group comparison.

### 2.4. Shotgun Analysis

For shot gun sequencing of samples, total DNA was isolated according to standard procedures (see Appendix A for more information). Library preparation was performed with 200 ng of total DNA with the NEBNext Ultra II FS DNA Library Prep Kit for Illumina (New England BioLabs, Frankfurt, Germany) according to the manufacturer’s instructions. Unique dual index primers (New England BioLabs, Frankfurt, Germany) were used for indexing, samples were pooled in equal ratios and a 4 nM pool was sequenced on a MiSeq desktop sequencer (Illumina, Eindhoven, Netherlands) with v3 600 cycles chemistry according to manufacturer’ instructions with 5% PhiX. FASTQ files were used for data analysis after demultiplexing. Raw sequence data were quality trimmed with Trim Galore (https://github.com/FelixKrueger/TrimGalore, accessed on 21 April 2021) dereplicated with Vsearch (https://github.com/torognes/vsearch, accessed on 21 April 2021) and finally analyzed using MetaPhlAn2 (https://github.com/biobakery/metaphlan2, accessed on 21 April 2021) and HUMAnN2 (https://github.com/biobakery/humann, accessed on 21 April 2021). All analysis steps were performed on the Medical University Graz private Galaxy instance (https://galaxy.medunigraz.at, accessed on 21 April 2021) running on a HPC infrastructure (MedBioNode, Graz, Austria).

### 2.5. Metabolomics

Cell free supernatant samples were analyzed using liquid chromatography–high resolution mass spectrometry. Detailed descriptions of the methods are listed in the Appendix A. After overnight metabolite extraction using cold methanol, samples were measured in duplicate in a stratified randomized sequence in one run with a Vanquish ultrahigh performance liquid chromatography (UHPLC) system coupled to a Qexactive mass spectrometer (Thermo Fisher Scientific) as described previously [38].

Raw data was converted into mzXML (msConvert, ProteoWizard Toolkit v3.0.5) and PeakScout (developed by Joanneum Research, Graz, Austria; [39]) was used to identify known metabolites using a reference list containing accurate mass and retention times.

Detected metabolites were quality controlled as described previously [40,41] using The Information Bus Company (TIBCO) Spotfire (v7.5.0, TIBCO, Palo Alto, CA, USA) and graded into two classes (I) suitable for multivariate and univariate analysis (MVA_UVA) and (II) suitable for univariate analysis (UVA). Technical variability was assessed calculating the relative standard deviation (RSD) of the quality control (QC) samples. A subsequent unsupervised dimensionality reduction analysis was performed using t-Distributed Stochastic Neighbor Embedding (t-SNE) to assess clustering of QCs and sample replicates. A detailed description can be found in the Appendix A. Correlations between metabolites and inoculation time, and metabolites and relative bacterial abundance were investigated.

### 2.6. In Vivo Murine Application

For in vivo application centrifuged and filtrated supernatant or MRS culture medium were used. The supernatant (production step 1 or 2) was thawed once for aliquoting to 1.5 mL Eppendorf vials and then stored at −80 °C. BALB/c mice (*n* = 20) were obtained at an age of 7 weeks from the Center for Biomedical Research of the Medical University of Vienna, Austria as one batch of littermates for microbiome testing. After delivery and an acclimatization period of two weeks, mice were split forming two equal groups (*n* = 10 each) with comparable body weight distribution. Animal experiments were approved by the veterinary board (BMBWF-66.010/0153-V/3b/2019). Mice were kept single-housed in individually ventilated cages under specific pathogen free conditions, a 12 h light-dark cycle and free access to chow and water at all times. After acclimatization, mice underwent a daily gavage with supernatant in the intervention group or MRS medium in the control group. Mice were gavage fed the 1:4 diluted supernatant (cultured for 48 h) or a 1:4 diluted MRS medium in the first week to check for possible side effects. Since there were no obvious negative reactions, mice were gavage-fed with the pure supernatant of production step 1 or pure MRS medium for 4 weeks. A stool sample was taken, and gavage feeding was paused for 4 weeks. Thereafter, the animals were gavage fed with the pure supernatant of production step 2 (cultured for 196 h) or culture medium for another 4 weeks and then euthanized (after 13 weeks with a total of 9 weeks of treatment). At euthanasia, a stool sample was collected and stored at −80 °C until 16S rRNA based microbiome analysis.

### 2.7. 16S rRNA Based Fecal Microbiome Analysis

The microbiome analysis was conducted as previously described [42]. For detailed information on sample preparation, measurement and analysis please see the Appendix A.

### 2.8. Statistics

Data was managed with Microsoft Excel 2016^®^ (Microsoft Corporation, Redmond, WA, USA) spreadsheets. Due to the small sample size normal distribution could not be assumed. Statistical analysis was conducted with SPSS 26.0^®^ (IBM Corporation, Armonk, NY, USA). Graphical workup was conducted with GraphPad Prism 9^®^ (GraphPad, San Diego, CA, USA). For heatmap analysis relevant results for gene families were normed to the maximum for each bacterium. The heatmaps were then drawn with the heatmap function of gplots package version 3.1.1 (gplot2, Elegant Graphics for Data Analysis, New York, NY, USA, https://ggplot2/tidyverse.org, accessed on 31 January 2022) for Rstudio^®^ version 1.4.1106 (R foundation for statistical computing, Vienna, Austria, https://R-project.org/, accessed on 31 January 2022). A Spearman-Rho analysis was conducted to detect significant correlations between markers obtained from different analyses. A *p* < 0.05 was considered statistically significant. Metabolite and inoculation time correlation graphs, as well as t-SNE plots were generated using the Orange Data Mining Toolbox v3.31.0 (Orange, The University of Lubljana, Lubljana, Slovenia, https://orangedatamining.com/, accessed on 25 January 2022).

## 3. Results

### 3.1. Bacterial Culture and Susceptibility Testing

The bacterial culture showed an initial increase of the optical density (OD) at 600 nm followed by a steady state with a final slight increase between 168 and 192 h (Figure 1). The CFU count increased up to 120 h followed by a decrease towards the end of the experiment.

Resistance testing revealed a higher antimicrobial effect of the postbiotic supernatant after 48 compared to 196 h (Figure 2). The postbiotic supernatant had no effect against anaerobic bacteria. Supernatant of both time points was reactive against *S. epidermidis* showing about half the reactivity at 196 compared to 48 h. Buffering reduced the effect of the 48 h supernatant against *L. monocytogenes*, *P. aeruginosae*, and *E. faecium*. Only the effect against *C. albicans* was negatively affected by cooking and proteinase K treatment.

### 3.2. Shotgun Analysis

The shotgun analysis did not allow us to discriminate between the two strains of *L. acidophilus* and between *B. longum* and *B. lactis* which were therefore combined for further analysis. A time-dependent decrease of the abundances of *L. salivarius*, *L. paracasei*, *E. faecium* and *B. longum/lactis* was noted. In contrast, *L. plantarum* showed an increase over the culture time. *L. rhamnosus* increased peaking at 96 h and then decreased again. *L. acidophilus* remained more or less constant (Figure 3).

In the pathway analysis, *E. faecium* presented the highest number of genes changing their expression over the culture time. This was followed by *L. paracasei*, *B. longum/lactis*, *L. rhamnosus*, *L. salivarius*, *L. plantarum*, *L. acidophilus* and *B. bifidum* (Figure 3). The full set of heatmaps can be retrieved from Appendix A. The bacteria of the co-culture differed in the special distribution regarding their gene activities. While *E. faecium*, *L. paracasei* had the highest expression at the beginning of the culture, *L. rhamnosus* and *L. salivarius* showed a peak at 96 h and *L. plantarum* at the end of the culture period at 196 h. *B. longum/lactis* had an initial peak followed by a plateau and a second peak at 48 h followed by a fast decrease in gene activity. *L. acidophilus* and *B. bifidum* revealed no clear pattern in view of culture time. The table in Figure 3 depicts the gene activities of the different bacteria allocated to different pathways according to MetaCyc (https://metacyc.org) (accessed on 15 January 2022).

### 3.3. VOC Analysis

A total of 36 substances could be identified in GC-MS. Two of these (propanal and pentane) could be attributed to room air contamination. 24 of these substances showed a significant correlation with the relative abundance of one or more of the probiotic bacteria over the different time points. The substances and their correlation with the relative bacterial abundance are displayed in Figure 4.

### 3.4. Metabolomics

After quality control, 122 metabolites were identified. MVA_UVA substances (98) were predominantly allocated to the groups amino acids/peptides/metabolites and nucleotides/metabolites, while UVA substances (24) belonged to the groups fatty acids and metabolites, carbohydrates and conjugates as well as pharmaceuticals and xenobiotics (Figure 5). Technical variability was excellent, with a median RSD of 4.4% in the QC samples for the 98 MVA_UVA metabolites. The good quality of the measurements was visually confirmed using t-SNE, showing a compact clustering of the QCs, and replicates overlapping in most cases.

The correlation analysis revealed the following: fumaric acid (R 0.75), malic acid (R 0.73), aspartic acid (R 0.66), cytidine monophosphate (R 0.66) and orotidine (R 0.64) increased with culture time while phosphoserine (R −0.84), creatine (R −0.80), panthotenic acid (R −0.75), tryptophan (R −0.74), and 9,3-methyl−2-oxovaleric acid (R −0.74) decreased. A detailed overview of the underlying pathways is given in Appendix A. The correlation analysis between the 98 metabolites and the relative bacterial abundance over the culture time revealed significant correlations in 25 cases (Figure 5). Of all bacteria, *L. paracasei* correlated with the metabolites to the highest extent, while *B. bifidum* had no significant correlations at all.

### 3.5. Effect of the Supernatant on the Fecal Murine Microbiome

Of 20 mice 1 in the control group had to be euthanized due to aspiration leaving 19 (*n* = 10 supernatant group and *n* = 9 control group) for microbiome analysis.

Neither alpha nor beta diversity markers were significantly different between mice fed with the postbiotic supernatant and culture medium at both time points (48 h and 196 h) (Figure 6). There were no significant differences at the phylum or the family levels (Appendix A). Gavage with the postbiotic supernatant after 48 h was associated with lower abundances of *Lachnospiraceae_FCS020_group* in Linear discriminant analysis effect size (LEfSe) and analysis of composition of microbiomes (ANCOM) analysis. LEfSe additionally revealed lower abundances of *Rikenellaceae_RC9_group* and higher abundances of *Anaeroplasma_uncultured* in the group that received the postbiotic supernatant (Figure 7). The postbiotic supernatant after 196 h of culture time was associated with a higher number of altered taxa in the murine fecal microbiome. *Ruminoclostridium_5_uncultured_Clostridium_bacterium* was increased in ANCOM and LEfSe analysis. Furthermore, LEfSe showed a decrease of *Lachnoclostridium_Dorea_sp_52* and *Ruminoclostridium_6_uncultured* as well as an increase of *Anaeroplasma_uncultured*, *Odoribacter_uncultured*, *Tyzerella_uncultured* and *Faecalibacterium prausnizii* associated with the postbiotic supernatant.

## 4. Discussion

Probiotic therapy aiming at a diverse and intact microbiome has gained increasing acceptance in health and disease in the past years. OMNi BiOTiC^®^ AAD10 combines 10 different probiotic species in a commercially available dietary supplement. Although there is plenty of information about the antimicrobial activity and the effect on the intestinal microbiome of single species, little is known about growth, gene expression, and metabolic activity of a combination of different probiotic bacteria when combined in a co-culture. Despite the beneficial effects of probiotics in selected cases the administration of live bacteria is discussed critically due to potentially harmful effects on the patients [23,24]. In these cases, health promoting alternatives are warranted [25]. Among these, postbiotics may be emerging as important microorganism-derived tool to promote health in the future [1]. Consequently, we were interested in the antimicrobial effect and composition of a culture supernatant from OMNi BiOTiC^®^ AAD10 and its effects on the fecal microbiome.

The timepoints of harvesting were chosen at 48 h to get an impression of the early phase of the culture and since other research groups investigating the susceptibility of supernatants reported the highest reactivity at this time point previously [43]. The culture was then continued to 196 h to investigate the interaction between species under competitive conditions in a long culture.

In our experiments, the 10 different probiotic strains were co-cultured in the same flask resembling one niche. We observed changes of the relative abundances of the probiotic strains with a decrease over time of *L. salivarius*, *L. paracasei* and *E. faecium* combined with an increase over time of *L. plantarum*. *L. rhamnosus* had a peak abundance at 96 h. *B. longum/lactis*, which could not be discriminated in this investigation (as explained above), had an initial peak followed by a plateau, a second peak at 48 h and decrease thereafter. The reason for the different growth behavior in this niche may be attributed to changes in the niche’s conditions (consumptions of space and nutrient resources, altered chemical composition as a consequence of metabolic or catabolic pathways or the release of bioactive macromolecules) [44].

The untargeted metabolome analysis of the culture supernatant revealed a correlation between several metabolites and the culture time. Especially the increase of fumaric and malic acid may have an impact on the different growth patterns as addition of organic acids may inhibit the growth of various bacteria in culture. In detail, malic acid effectively inhibited growth of *Listeria monocytogenes*, *Escherichia coli*, *Salmonella typhimurium* [45] and *Shigella flexneri* [46]. Similarly, fumaric acid exhibited clear antimicrobial activity against *Campylobacter jejuni* [47]. The impact and the underlying mechanisms of single metabolic products on the growth behavior of different probiotic strains, however, remain unclear at present.

Besides environmental changes a direct interaction between bacteria can be assumed. Although the underlying molecular mechanisms for the interaction between the different bacterial strains are very complex, quorum sensing—A bacterial cell-to-cell communication process—Seems to play a major role (reviewed in [48]). Quorum sensing may induce bacteria to produce bacteriocins or/and other antimicrobial molecules aiding bacteria in the defense of their habitat. In this context, the co-culture of different bacteria may have beneficial effects triggering the induction of ribosomally and non-ribosomally produced secondary metabolites [48]. A co-culture of nisin-producing *Lactococcus lactis* with *Saccharomyces cerevisiae,* for instance, increased the nisin production by 85% [49].

The targeted metabolome analysis was not focused on substances held responsible for the modulation of quorum sensing (acetylated homoserine lactons, small peptides or autoinducer-2) [29], thus not allowing for a conclusion regarding these metabolites.

Regarding the bacterial gene expression in the shotgun analysis the vast majority of underlying pathways belonged to the sectors nucleoside/nucleotide biosynthesis, energy production (fermentation, glycolysis) and protein/amino acid synthesis. Except *L. salivarius* (with a peak activity at 96 h despite decreasing relative abundance) the gene expression of the bacteria resembled their relative abundance. Consequently, the untargeted metabolome analysis also revealed nucleotides and their metabolites, amino acids/peptides and energy metabolism as most prevalent classes with specific substances correlating with the relative bacterial abundance. Of all probiotic bacteria in this study, *L. paracasei* had the highest number of correlations while *B. bifidum* had none. The reason for this finding may lie in a high metabolic activity of *L. paracasei* together with low activity and low relative abundance of *B. bifidum* in this co-culture experiment. The volatile organic compounds consumed/emitted by the bacterial culture predominantly belonged to the classes esters, ketones, and aldehydes. As individual substances may be produced by many different species it is difficult to draw direct conclusions on bacterial activity.

The postbiotic supernatant produced in our study showed in vitro antimicrobial activity against *S. epidermidis, L. monozytogenes, P. aeruginosae, E. faecium, Str. Agalactiae*, and *C. albicans.* The antimicrobial activity depended on the culture time and was more pronounced after 48 h than after 196 h. We chose 48 h as first time point for harvesting, since previous investigations of monocultures demonstrated a high antimicrobial activity of the supernatant at that time point [43]. To the best of our knowledge there are presently no other reports about the antimicrobial activity of longer culture periods. At this later time point the activity against *L. monocytogenes*, *P. aeruginosae*, *E. faecium*, and *C. albicans* was lost. However, the supernatant at 196 h inhibited the growth of *Str. agalactiae* which had not been observed before.

There are several publications regarding the antimicrobial activity of cell free supernatants (CFS) [50,51,52]. However, most of them deal with either monocultures [50,51,52] or mix the CFS of different bacteria after culture [53]. Danilova et al., for instance, demonstrated antimicrobial activity of *L. plantarum* culture supernatant against *E. coli, P. aeruginosa, S. pyogenes* and *S. aureus* [43]. In a detail they could attribute the antimicrobial effect to the low-molecular-weight, but not to the high-molecular-weight fraction of their supernatant. Compared to their data we also noticed an antimicrobial effect against *P. aeruginosae* but not against *E. coli* and *S. aureus*. As we did not differ between low- and high-molecular weight fractions, we could not attribute the antimicrobial effects to either fraction. Regarding the effect of a combination of different probiotic species Fredua-Agyeman et. al. demonstrated an inhibitory effect of a mixture of probiotic culture supernatants of three different probiotic species (*L. acidophilus, B. lactis* and *B. bifidum*) against *P. aeruginosa* [53]. As different species will influence each other during culture in the same niche, and induction of various bacteriocins can be triggered by co-culture mixing of supernatants will probably differ from the CFS of a co-culture. Antifungal activities similar to our data have also been previously reported for *Lactobacillus* strains [54] and CFS of *Pediococcus acidilactici HW01* [55].

The antibacterial activity of our postbiotic supernatant was neither influenced by subjection to proteinase K nor by cooking paralleling other reports in the literature [43]. Both treatments, however, mitigated its antifungal activity. Hence, the effect against *C. albicans*, but not the antibacterial activity seems to rely on some sort of peptide or protein. As the supernatant analysis were not targeted to bacteriocins, the reason for the antimicrobial activity of our postbiotic supernatant has still to be elucidated. Maybe future analysis with a targeted approach for these substances responsible for quorum sensing and different bacteriocins can help to understand the different growth patterns and the antimicrobial effect of our co-culture.

In special circumstances, as for instance immunocompromised patients, the administration of live bacteria in form of probiotics maybe critical [24]. In these cases, bacteriocins or postbiotic supernatant could be a useful alternative [27]. In this regard postbiotics from *Lactobacilli* have demonstrated beneficial immunomodulatory, anti-tumor, antimicrobial and barrier-preserving effects on the host [56]. Previous studies have shown an impact of postbiotics on the intestinal microbiome in various models. One study with a *Lactobacillus* based postbiotic in rainbow trout revealed a significant increase of diversity (Shannon) and richness (Chao1) as well as altered abundances with a decrease of Fusobacteria and an increase of Tenericutes, Spirochaetes and Bacteroides under postbiotic treatment in fish fed with postbiotics [57]. An investigation of postbiotic supplementation in suckling rats also revealed an increase in richness and a lower abundance of *Chitinophagaceae* in the postbiotic compared to the reference group [58]. Mice fed with a heat treated postbiotic from *L. fermentum* and *L. delbrueckii* showed a reduction of *Turicibacter*, *Clostridium sensu strictu* and *Dorea* [59]. The application of the same probiotic in a fecal fermentation model of the human gut revealed a heat and enzyme stable bifidogenic effect [24]. In our mouse model postbiotic supernatant from OMNi BiOTiC^®^ AAD10 did neither affect alpha nor beta diversity, which is in contrast to other studies [57,58]. At the genus level, we found a differential impact of CFS after 48 h and 196 h of culture time. After 48 h *Anaeroplasma* increased and *Lachnospiraceae_FCS020_group* and *Rikenellaceae_RC9_group* decreased with possible beneficial effects for the host [60]. The increase of potentially health promoting *Anaeroplasma* remained present when administering postbiotic supernatant after 196 h of culture. Additionally, there was an increase of *Odoribacter*, *Ruminoclostridium_5*, *Tyzzerella* and *Faecalibacterium (F.) prausnitzii* together with a decrease of *Dorea* and *Ruminoclostridium_6*. Of these, especially the increase of *F. prausnitzii* has been associated with beneficial aspects as production of SCFAs and anti-inflammatory effects [61,62]. *Dorea* has been associated with non-alcoholic fatty liver disease [63,64]. Consequently, a reduction of *Dorea* may also be beneficial for the host. The different effects of postbiotic CFS observed in various studies may be related to the different models (fish, rats, mice, fermenter cultures) and the different classes of postbiotics (cell free supernatant, exopolysaccharides, enzymes, cell wall fragments, bacterial lysates, SCFAs, bacteriocins [27]) investigated.

The limitations of the present study are that OMNi BiOTIC^®^ AAD10 is a food supplement and not registered as a drug. Despite rigorous quality controls by the manufacturer this may have an impact on the quality. Another limitation is the characterization of the CFS culture, which was only performed once. The shotgun, VOC and metabolomic data presented relies on two samples per time point of this culture. Consequently, we refrained from performing any advanced statistics. Although the presented data have to be verified with multiple repetitions to generate statistically sound data in the future (especially when conducting targeted approaches in search of bacteriocins), we still provide the first insights into the processes of a probiotic co-culture. Storage at −80 °C for only short periods without repeated freezing and thawing should ascertain stability of the supernatant. Repeated metabolomics analysis over the time would have been required as prove. However, this was not within the scope of this investigation. The susceptibility testing was also based on one culture but was repeated five times for each species tested and the antimicrobial activity of the postbiotic supernatant is in close relation to data previously published. While the in vitro susceptibility testing gives first insights on the antimicrobial activity of the supernatant its reactivity maybe different in vivo. The fecal microbiome only presents a limited information of the total intestinal microbiome. Every intestinal segment has its own microbiome and there are even differences between luminal and mucosal microbial compositions [65]. Although other intestinal compartments have not been addressed, here the fecal microbiome allows for the first insights into the effect of our supernatant. The nomenclature of the probiotic bacteria was applied as applicable before the changes made in 2020 [66], since these names are still used by the manufacturer of OMNi BiOTiC^®^ AAD10.

## 5. Conclusions

In this study we could characterize the co-culture of 10 probiotic bacteria contained in the food supplement OMNi BiOTiC^®^ AAD10. We demonstrated a time dependent change of the relative abundance and gene expression levels of probiotic bacteria in co-culture. Gene expressions could be predominantly allocated to the pathways of nucleotide and amino acid biosynthesis, which was in accordance with findings from the metabolome analysis. The postbiotic supernatant of OMNi BiOTiC^®^ AAD10 had positive antibacterial and antifungal effects in vitro and promoted the growth of beneficial bacteria in the murine model. Future studies will have to determine the effects of OMNi BiOTiC^®^ AAD10 derived supernatant for the human use in health and disease.

## Figures and Tables

**Figure 1 nutrients-14-01194-f001:**
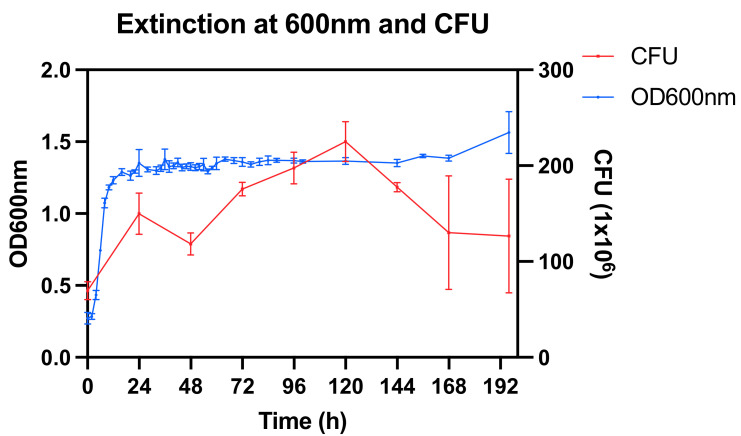
Optical density (OD) at 600 nm and colony forming units (CFU) at 10^6^ dilution.

**Figure 2 nutrients-14-01194-f002:**
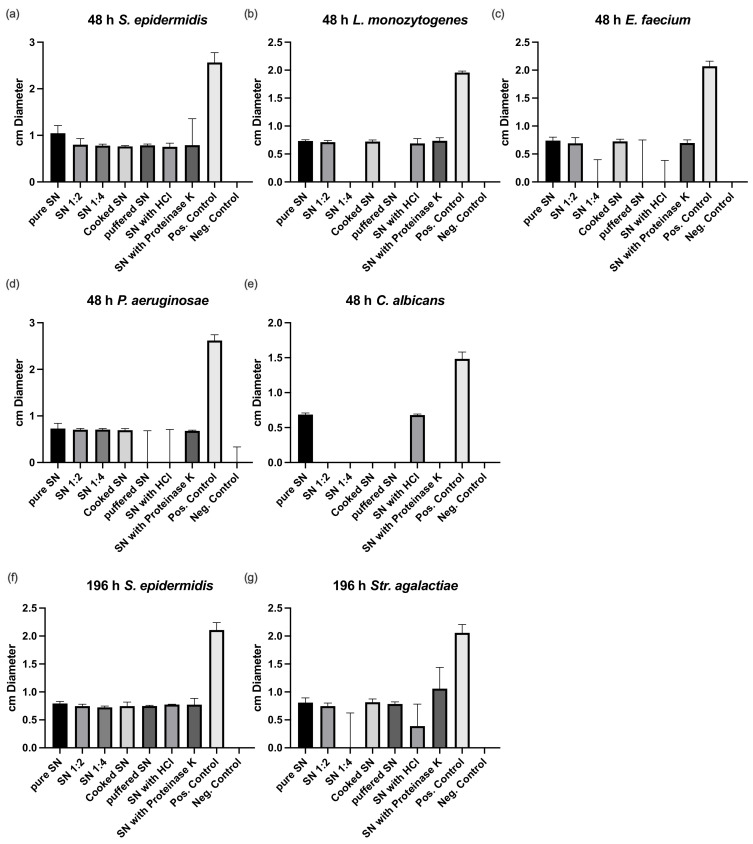
Susceptibility testing of culture supernatant after 48 (**a**–**e**) and 196 h of culture (**f**,**g**). SN, supernatant; HCl, hydrochloric acid. Pos, positive; Neg, negative.

**Figure 3 nutrients-14-01194-f003:**
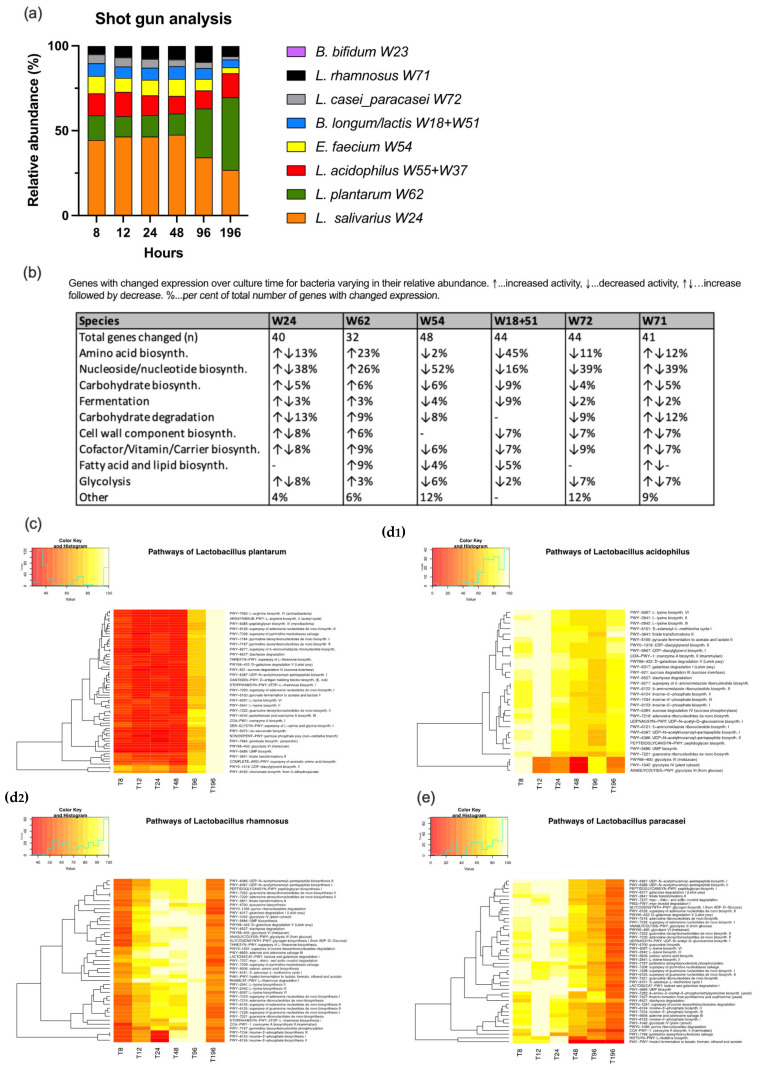
Shotgun analysis. Relative abundances (**a**); gene expressions of bacteria allocated to different pathways and change over time (**b**); heatmaps of selected bacteria (**c**–**e**). 0% of maximum colored in red and 100% in yellow.

**Figure 4 nutrients-14-01194-f004:**
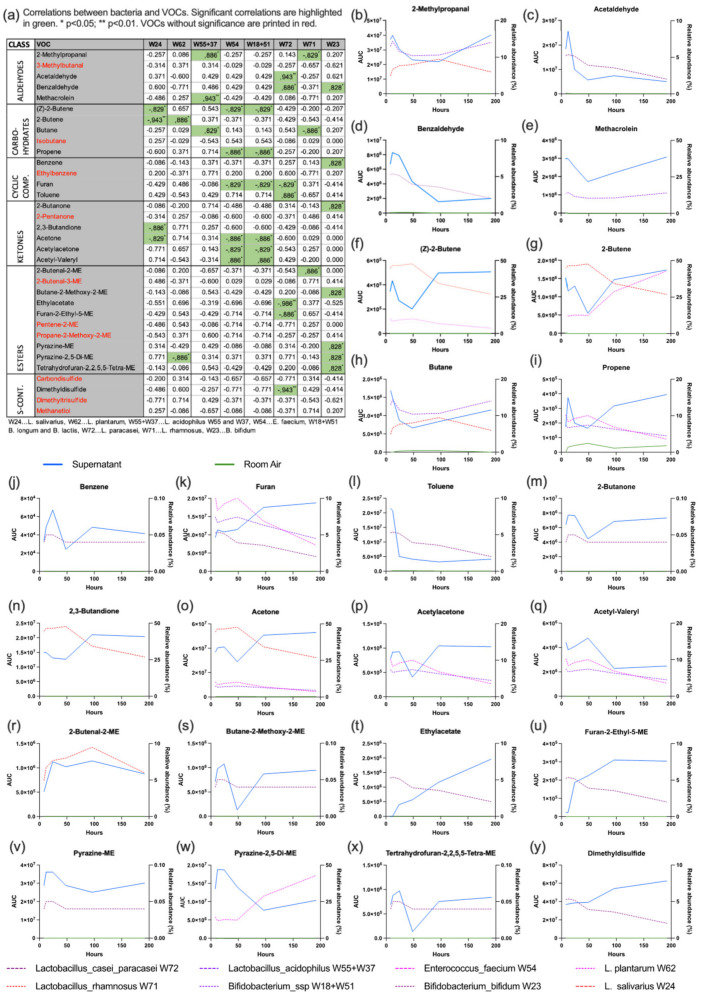
VOC analysis. The table (**a**) gives an overview of all VOCs and their correlation with the relative abundance of probiotic bacteria. The graphs (**b**–**y**) show the profile of the VOC (blue solid line), the room air (green solid line) and the relative abundance of the correlating probiotic bacterium (dashed lines).

**Figure 5 nutrients-14-01194-f005:**
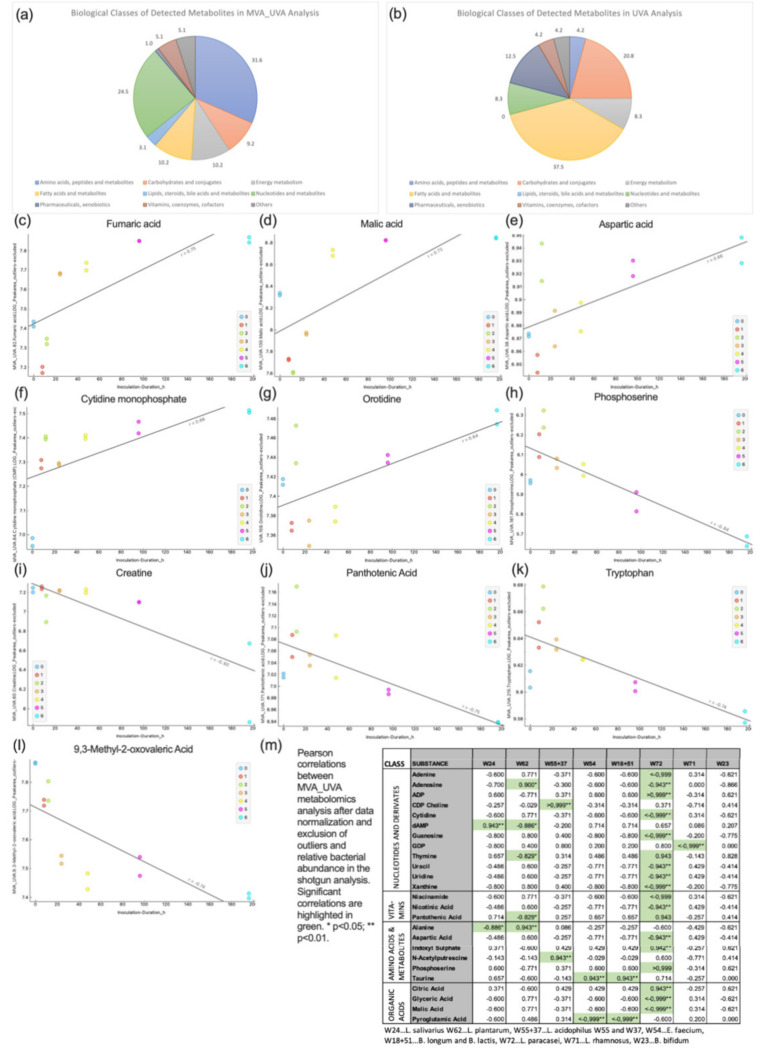
Metabolomics analysis; biological classes of metabolites detected in MVA_UVA (**a**) and UVA (**b**) analysis; substances with positive (**a**–**g**) and negative correlation (**h**–**l**) with inoculation time and correlation of metabolites with the relative abundance of probiotic bacteria (only metabolites with significant correlations are displayed) (**m**) correlation between metabolites and relative bacterial abundance.

**Figure 6 nutrients-14-01194-f006:**
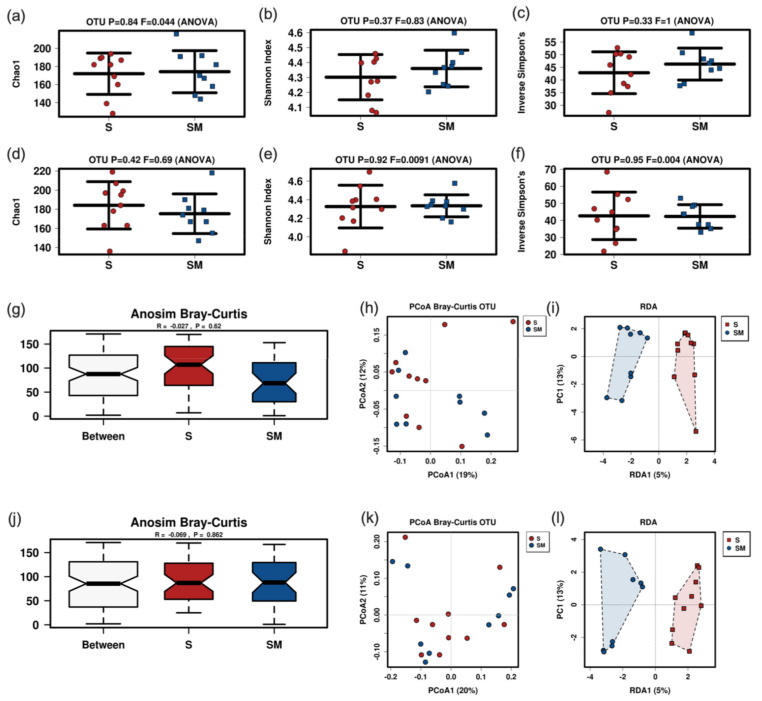
Results of murine fecal sample microbiome analysis after gavage of supernatant or control. Alpha diversity markers of supernatant harvested after 48 h (rarefication 14,420 reads) (**a**–**c**) or 196 h (rarefication 14,111 reads) culture time (**d**–**f**). Anosim Bray-Curtis and PCoA Bray Curtis plots for 48 h (**g**,**h**) and 196 h (**j**,**k**) supernatant. RDA for 48 h supernatant (**i**): *p* = 0.893; F 0.85; Var 18.34. RDA for 196 h supernatant (**l**): *p* = 0.699; F 0.92; var 19.3. S, supernatant; SM, medium group.

**Figure 7 nutrients-14-01194-f007:**
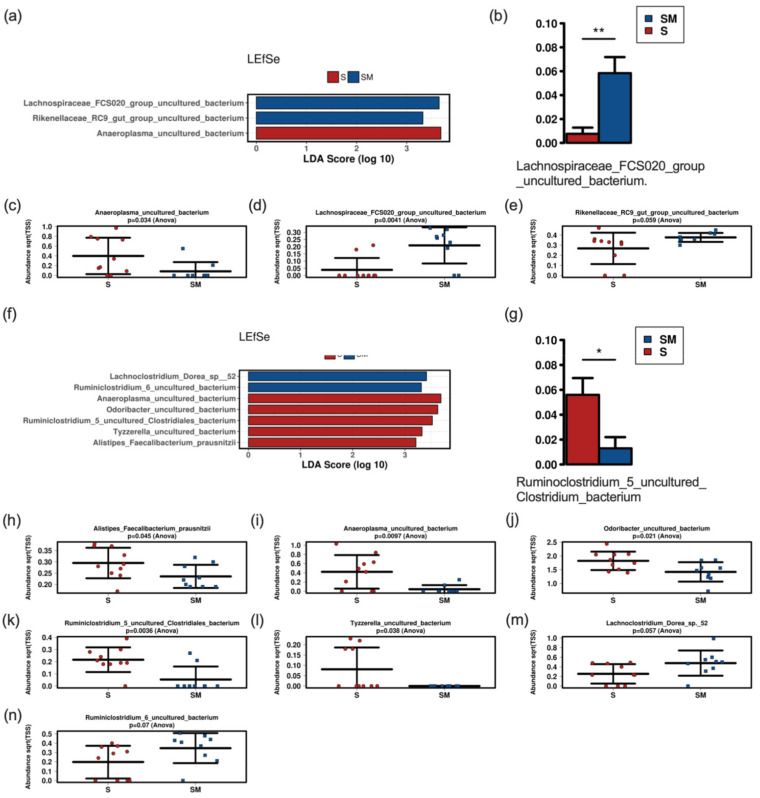
LEfSe (**a**,**f**) and ANCOM (**b**,**g**) analysis with bar charts (**c**–**e**,**h**–**n**) of fecal samples of mice treated with the 48 h (**a**–**e**) and the 196 h (**f**–**n**) postbiotic supernatant versus mice treated with culture medium. S, supernatant; SM, medium control. * significant difference (*p* < 0.05), ** significant difference (*p* < 0.01).

## Data Availability

The data presented in this study are deposited in the European Nucleotide Archive (https://www.ebi.ac.uk/ena/browser/home, accessed on 8 March 2022), accession number PRJEB50450.

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
