# Peer review of "Insights into the Composition of a Co-Culture of 10 Probiotic Strains (OMNi BiOTiC® AAD10) and Effects of Its Postbiotic Culture Supernatant"

_nutrients, 2022, doi:10.3390/nu14061194_

Round 1
Reviewer 1 Report
The use of probiotic bacteria, in many cases combined with prebiotic substrates for growth and functional molecule or structure production, has received very considerable recent attention. Less so is the use of as the authors all postbiotics which can be culture supernatants or processed dead bacteria. There is no a priori consideration that postbiotics would be inferior. Infact in some patients under certain conditions, use of live bacteria may not be desirable. Use of probiotic mycotic species is considered carefully in particular for immunocompromised, including the elderly and young as giving mycotic species may have the possibility of dire consequences. This concept is very important for this project, but is discussed to a modest degree not in the Introduction but only in the Discussion. There is little presentation why someone should not eat probiotic bacteria or mycotic organisms, what harmful effects are possible in some people? Of note probiotic bacteria have been studied in very young children and the very elderly and in the vast majority of occasions can be safely used. This is not to detract from the present and other studies as identifying lead compounds and this is still valid, but not clearly presented as a concept either.
One set of information that will be returned to later but needs early mention is that the OMNI Biotic AAD10 is available for purchase and consumption online and this may be so because it is a food product and not in many countries regulated that same method as a identified chemical or drug. Of note after extensive searching there is no basic science reporting on this probiotic mixture. It is challenging to find any clinical reports of substance on this probiotic mixture, all reports are anecdoctal and present no numbers and methodology. The characterization of the metabolites in the AAD10 culture supernatants is interesting and important and COULD provide leads for potential treatment agents. However, the manner that the studies are constructed in and very modest experimentation presented limits the value of the studies.
Additional major points
1) One of the major items not included in the manuscript is why one would want to use postbiotic rather than feeding a host with a mixture of probiotics? One of the longest developed, most widely used probiotic is VSL#3. Work has been done with this product why the combination of many bacteria and in VSL#3 also one mycotic, why is the mixture important. This is a concept that is missing from the present manuscript. There are concepts that are incompletely presented or not at all. Growth of one bacteria can effect the composition and structures and secreted products of other bacteria, thus a population can act differently than the sum of potential actions of all individual species. Second is the concept of synergy of action of multiple components, a central concept to natural product medicine. No rationale presented why these 10 organism types are used together. There is no information presented on the nature of the individual bacteria types and their characteristics that make them good probiotics. There is no information on how these bacteria communicate with each other and effects their functions.
The authors should do a Google search using the term postbiotic and they will find not only advertisements for specific bacterial metabolites, but also discussion on what are the best postbiotics even in such publications as US News and World Reports https://health.usnews.com/health-care/patient-advice/articles/what-are-postbiotics. There are many many references to health oriented websites that are opinionated and not founded in scientific investigations, but there are many sources available.
2) Background and literature citation is not as complete and extensive as it should be. Citation number 1 is a Nature Reviews which is one of the more recent important to cite. The authors should read this citation carefully and expand on the concepts of why postbiotics may be so appealing including advantages and disadvantages. The Nature Reviews introduces other concepts that the reader of the present manuscript should be informed of without needing to read previous reviews in this field. There is almost no original literature cited, basic research or clinical trials and this is also a deficit in the background presented in this manuscript.
3) In vivo data on the microbioome demonstrates a modest effect and there is no way to determine whether this is an important functional effect. Studies on the role of the intestinal microbiome have demonstrated the role of the bacteria in organ functions of intestine, liver, brain, kidney, immune, respiratory and nearly all systems. Unfortunately to date, the studies are correlative and do not prove the role of the bacteria. It is often neglected what components or structures or secreted products of the bacterial population are involved in the effect. Manners to specifically intervene are sorely needed to provide causative data, germ free mice may provide some evidence with conventionalization, but is not a great model. There is always some pathophysiologic condition involved, some manner to perturb first some function and next to alter the microbiome and see what effect this has on the altered pathophysiology. The present studies have no organismal function of study, solely give the postbiotic and look at changes. This is not a great deal of information and may not be very important. It should be noted that in the last line of the abstract the word beneficial is used to describe some bacteria, but there is no experimentation and model used to test this claim.
4) In vivo antimicrobial activity may be very different as conditions in the small and large intestine are very different from the in vitro anti microbial assay condition used. Included in the Google and PubMed search results are a number of studies on postbiotics used for in vivo antimicrobial activity. There is data within these studies that demonstrate that the in vivo antimicrobial activity is not the same as the in vitro studies, but this is not discussed. There is also no explanation for differences in sensitivity to the antimicrobial actions and in particular why anaerobic bacteria might be unaffected.
Minor Points
1) Why were the two time points chosen? It is noted that some activities are greater at 48 versus 196 hours but there is no comment made. One concern is therefore the stability of some factors in the culture supernatant and this is not addressed.
2) The MRS broth and growth conditions may be good or even optimal for some bacteria in the mixture, but not for others. Each bacteria could demonstrate a different temporal course of abundance but this is not readily apparent from the data as presented.
3) The authors are well informed about bacteriocins and there is an extensive literature on these. It would appear from how the manuscript is written that the present studies are more novel in study of bacterial culture supernatants as antimicrobials, but a Google search identifies dozens of publications. Antimicrobial peptides and other molecules from bacteria or mycotic organisms are very important and will be useful and have been studied in greater detail and depth than this action studied in the present manuscript. The mechanisms of antimicrobial activity have in a number of cases been studied. https://www.google.com/search?q=bactericions&rlz=1C5CHFA_enUS903US903&ei=UEYHYp_VCYayggeAvr2oDg&start=10&sa=N&ved=2ahUKEwifvZKFuPn1AhUGmeAKHQBfD-UQ8tMDegQIARA8&biw=1331&bih=596&dpr=1
or for mycotic derivation
https://www.google.com/search?q=mycotic+derived+antimicrobial+substances&rlz=1C5CHFA_enUS903US903&biw=1331&bih=596&ei=c0YHYveFF6y3ggfMgrWYDw&oq=mycotic+derived+antimicrobial+su&gs_lcp=Cgdnd3Mtd2l6EAEYADIFCCEQoAEyBQghEKABMgUIIRCgATIFCCEQqwIyBQghEKsCMgUIIRCrAjoHCAAQRxCwAzoHCAAQsQMQDToECAAQDToHCAAQsQMQCjoKCAAQsQMQgwEQCjoECAAQCjoICAAQgAQQsQM6BQgAEIAEOgcILhCABBAKOggILhCABBCxAzoLCAAQgAQQsQMQgwE6CwguEMcBEK8BEJECOgUIABCRAjoLCC4QgAQQsQMQ1AI6EQguEIAEELEDEMcBENEDENQCOg4ILhCxAxDHARCjAhCRAjoLCC4QxwEQ0QMQkQI6DgguEIAEELEDEMcBEKMCOhEILhCABBCxAxDHARCjAhDUAjoOCC4QgAQQsQMQgwEQ1AI6CwguEIAEEMcBEKMCOgsILhDHARCjAhCRAjoLCC4QgAQQsQMQgwE6BQguEIAEOhQIABDqAhC0AhCKAxC3AxDUAxDlAjoFCC4QkQI6CwguELEDEIMBENQCOggILhCxAxCDAToICAAQsQMQkQI6CAguEIAEENQCOgsIABCxAxCDARCRAjoICAAQsQMQgwE6CwguEIAEEMcBEK8BOgcIABCABBAKOgYIABAWEB46BQgAEIYDOgcIIRAKEKABOggIIRAWEB0QHkoECEEYAEoECEYYAFDjBFiMZGD7eWgHcAF4AIABiwGIAeMhkgEEMzkuOZgBAKABAbABCsgBCMABAQ&sclient=gws-wi
Author Response
The authors would like to extend their gratitude to the reviewers for their thorough work and their input, which greatly helped to improve this manuscript. In the following, please find our detailed responses to the comments of the 3 Reviewers.
REVIEWER 1:
The use of probiotic bacteria, in many cases combined with prebiotic substrates for growth and functional molecule or structure production, has received very considerable recent attention. Less so is the use of as the authors all postbiotics which can be culture supernatants or processed dead bacteria. There is no a priori consideration that postbiotics would be inferior. Infact in some patients under certain conditions, use of live bacteria may not be desirable. Use of probiotic mycotic species is considered carefully in particular for immunocompromised, including the elderly and young as giving mycotic species may have the possibility of dire consequences. This concept is very important for this project, but is discussed to a modest degree not in the Introduction but only in the Discussion. There is little presentation why someone should not eat probiotic bacteria or mycotic organisms, what harmful effects are possible in some people? Of note probiotic bacteria have been studied in very young children and the very elderly and in the vast majority of occasions can be safely used. This is not to detract from the present and other studies as identifying lead compounds and this is still valid, but not clearly presented as a concept either.
Response: Thank you very much for this comment. We agree that the manuscript benefits from an early discussion regarding possible problems with probiotics. We have modified the introduction section accordingly.
One set of information that will be returned to later but needs early mention is that the OMNI Biotic AAD10 is available for purchase and consumption online and this may be so because it is a food product and not in many countries regulated that same method as a identified chemical or drug. Of note after extensive searching there is no basic science reporting on this probiotic mixture. It is challenging to find any clinical reports of substance on this probiotic mixture, all reports are anecdoctal and present no numbers and methodology. The characterization of the metabolites in the AAD10 culture supernatants is interesting and important and COULD provide leads for potential treatment agents. However, the manner that the studies are constructed in and very modest experimentation presented limits the value of the studies.
Response: Thank you very much for this remark. We have included more information about OMNi BiOTiC® AAD10 as commercially available food supplement in the introduction and discussion section. We agree that the clinical and general scientific information in OTHER studies may be limited. This manuscript aims on providing basic data on this probiotic. Of course, this will not help to improve the quality previous studies published by other authors but should give some basic information to build on in future studies dealing with this agent.
1) One of the major items not included in the manuscript is why one would want to use postbiotic rather than feeding a host with a mixture of probiotics? One of the longest developed, most widely used probiotic is VSL#3. Work has been done with this product why the combination of many bacteria and in VSL#3 also one mycotic, why is the mixture important. This is a concept that is missing from the present manuscript. There are concepts that are incompletely presented or not at all. Growth of one bacteria can effect the composition and structures and secreted products of other bacteria, thus a population can act differently than the sum of potential actions of all individual species. Second is the concept of synergy of action of multiple components, a central concept to natural product medicine. No rationale presented why these 10 organism types are used together. There is no information presented on the nature of the individual bacteria types and their characteristics that make them good probiotics. There is no information on how these bacteria communicate with each other and effects their functions.
Response: Thank you very much for this comment. As you mentioned yourself in your introductory comment there may be situations, where probiotic administration can be discussed critically (as neonates, elderly, or immunocompromised patients) because of the risks of bacteremia, spread of antimicrobial resistance or altered long-term immune responses. We agree that mentioning these effects as rationale for the use of postbiotics is an important issue which has missed in our introduction. As commented above we have included this information now. As discussed in the aims section it was one of the aims of this study to gain first insights on the behavior of the 10 different probiotic strains used in OMNi BiOTiC AAD10. All bacteria contained in the food supplement are acknowledged probiotic bacteria. We have included more information about the strains contained in OMNi BiOTiC® AAD10 and some of their functions. A detailed work-up of all the literature available on the single strains would probably fill books. If the reviewer has further detailed demands which should be addressed in this manuscript we will be glad to do so in our next revision.
The authors should do a Google search using the term postbiotic and they will find not only advertisements for specific bacterial metabolites, but also discussion on what are the best postbiotics even in such publications as US News and World Reports https://health.usnews.com/health-care/patient-advice/articles/what-are-postbiotics. There are many many references to health oriented websites that are opinionated and not founded in scientific investigations, but there are many sources available.
Response: We thank the reviewer for this very valuable remark. Of course, there are many different sorts of postbiotics. In our literature search we like to stick to scientific databases as Pubmed rather than trivial search engines. It was not our desire to discuss the differences between different postbiotics, but to investigate the composition and effects of a specific postbiotic.
2) Background and literature citation is not as complete and extensive as it should be. Citation number 1 is a Nature Reviews which is one of the more recent important to cite. The authors should read this citation carefully and expand on the concepts of why postbiotics may be so appealing including advantages and disadvantages. The Nature Reviews introduces other concepts that the reader of the present manuscript should be informed of without needing to read previous reviews in this field. There is almost no original literature cited, basic research or clinical trials and this is also a deficit in the background presented in this manuscript.
Response: Thank you for this suggestion. We have included some information on the possible positive effects of postbiotics and some clinical data available in the introduction section of the revised manuscript.
3) In vivo data on the microbioome demonstrates a modest effect and there is no way to determine whether this is an important functional effect. Studies on the role of the intestinal microbiome have demonstrated the role of the bacteria in organ functions of intestine, liver, brain, kidney, immune, respiratory and nearly all systems. Unfortunately to date, the studies are correlative and do not prove the role of the bacteria. It is often neglected what components or structures or secreted products of the bacterial population are involved in the effect. Manners to specifically intervene are sorely needed to provide causative data, germ free mice may provide some evidence with conventionalization, but is not a great model. There is always some pathophysiologic condition involved, some manner to perturb first some function and next to alter the microbiome and see what effect this has on the altered pathophysiology. The present studies have no organismal function of study, solely give the postbiotic and look at changes. This is not a great deal of information and may not be very important. It should be noted that in the last line of the abstract the word beneficial is used to describe some bacteria, but there is no experimentation and model used to test this claim.
Response: We completely agree with the reviewer as to the possible effects of probiotics. A large proportion of their effects could be attributed to bacterial metabolites as secondary/tertiary bile acids, MAMPs or SCFAs. This further fuels the discussion as to the effect of pro- and postbiotics – is there a difference in the organism if you feed a probiotic or its postbiotic (including inactivated bacteria, bacterial fragments or solely metabolites)? With the present project we aimed to gain some first insights into this matter without claiming completeness. In future studies we aim to compare the effect of pro- and postbiotic in a “healthy” mouse model. Furthermore, we are of course interested in the effect of postbiotics in diseases and investigations are currently being conducted in cancer models. Including all this data however would exceed the scope of one original article by far. We have included a paragraph discussing this issue in the introduction section of the revised manuscript.
4) In vivo antimicrobial activity may be very different as conditions in the small and large intestine are very different from the in vitro anti microbial assay condition used. Included in the Google and PubMed search results are a number of studies on postbiotics used for in vivo antimicrobial activity. There is data within these studies that demonstrate that the in vivo antimicrobial activity is not the same as the in vitro studies, but this is not discussed. There is also no explanation for differences in sensitivity to the antimicrobial actions and in particular why anaerobic bacteria might be unaffected.
Response: We completely agree to the opinion of the reviewer. Of course, an in vitro test can only give first insights on possible effects in vivo. Nevertheless in vitro resistance testing is an important tool in gaining insights in reactivities of antibiotics and antimycotics in the clinical routine. We agree that an observed in vitro effect may not be observed in vivo. The gut alters its microbial composition within every segment. In the past we could demonstrate these effects in children and mice. We could even describe differences between mucosal and luminal microbial compositions throughout the gut. We also previously examined other postbiotics as reuterin system or reuterizyklin in vivo and in vitro. We thank the reviewer for providing more literature on this topic. We have included a paragraph discussing this issue in the study limitations section of the revised version of the manuscript. We agree that the term beneficial is inappropriate in the abstract as it requires mentioning of correspondent literature. The abstract has been modified accordingly.
Minor Points
1) Why were the two time points chosen? It is noted that some activities are greater at 48 versus 196 hours but there is no comment made. One concern is therefore the stability of some factors in the culture supernatant and this is not addressed.
Response: Thank you very much for this remark. Regarding the choice of time points we would like to refer to the following statement in the discussion section “The timepoints of harvesting were chosen at 48 h to get an impression of the early phase of the culture and because other research groups investigating the susceptibility of supernatants reported the highest reactivity at this time point previously [39]. The culture was then continued to 196 h to investigate the interaction between species under com-petitive conditions in a long culture. As to the stability: the supernatant was stored at -80°C in an ultrafreezer which should grant stability for most components. However, repeated tests over the time would have been necessary to ascertain its stability. This was not within the scope of this project. However, this is an important limitation and we have included a remark in the limitations section of the revised version of the manuscript.
2) The MRS broth and growth conditions may be good or even optimal for some bacteria in the mixture, but not for others. Each bacteria could demonstrate a different temporal course of abundance but this is not readily apparent from the data as presented.
Response: We agree, that MRS may be better for some bacteria in the probiotic mixture than for others. Experiments with different culture media were not within the scope of this study. As to the different temporal course of bacterial abundance we would like to refer to figure 3, panel a and the corresponding text in the beginning of subheading 3.2
3) The authors are well informed about bacteriocins and there is an extensive literature on these. It would appear from how the manuscript is written that the present studies are more novel in study of bacterial culture supernatants as antimicrobials, but a Google search identifies dozens of publications. Antimicrobial peptides and other molecules from bacteria or mycotic organisms are very important and will be useful and have been studied in greater detail and depth than this action studied in the present manuscript. The mechanisms of antimicrobial activity have in a number of cases been studied.
Response: Thank you for this remark and the links you provided. Of course, there are multitudes of publications regarding postbiotics and probiotics in google and professional databases as Pubmed. We never claimed to be the only ones investigating this issue. However, we are first to present insights into a probiotic co-culture of 10 different bacteria.
Reviewer 2 Report
This is a study aimed at gaining insights the composition of a co-culture of 10 bacteria and the effect of their postbiotic supernatant. Interestingly, a time-dependent decrease of the relative abundances and gene expression of L. salivarius, L. paracasei, E. faecium and B. longum/lactis and an increase of L. plantarum was observed. More important, the postbiotic supernatant had positive antibacterial and antifungal effects in vitro and promoted the growth of beneficial bacteria in vivo. This study demonstrates that it is possibile to fight microrganisms by using microrganisms and that we may act a strategy to make non culturable species growing-up. The manuscript is well written and the results of relevance.
Author Response
Response: thank you very much for this encouraging remark!
Reviewer 3 Report
This is an interesting and well written paper, however, although the authors write that Probiotic therapy aiming at a diverse and intact microbiome has gained increasing acceptance in health and disease in the past years it has also not been proven as effective as expected and has some downsides. These should be addressed in the Manuscript.
Since the product used is a dietary supplement and they do not undergo rigorous controls as drugs, was the product tested first for the number of live strains? If not this should be included in the limitations.
Also I am not sure how may the results of the study translate into clinical practice, can something be said about that? The susceptibility testing was based on one culture limits the possible translation of the results into clinical practice.
Author Response
The authors would like to extend their gratitude to the reviewers for their thorough work and their input, which greatly helped to improve this manuscript. In the following, please find our detailed responses to the comments of the 3 Reviewers.
REVIEWER 3:
This is an interesting and well written paper, however, although the authors write that Probiotic therapy aiming at a diverse and intact microbiome has gained increasing acceptance in health and disease in the past years it has also not been proven as effective as expected and has some downsides. These should be addressed in the Manuscript.
Response: We agree with the reviewer regarding possible downsides of probiotics. We have included a paragraph regarding adverse events under probiotic therapy in the introduction section of the revised version of the manuscript.
Since the product used is a dietary supplement and they do not undergo rigorous controls as drugs, was the product tested first for the number of live strains? If not this should be included in the limitations.
Response: Thank you for this comment. The product undergoes rigorous internal quality controls by the producer but is no drug but a food supplement. We agree that this should be mentioned in the study limitations section and have included a statement commenting on this topic. The number of live strains has been checked at the beginning of the culture and throughout the culture process (see methods section and figure 1).
Also I am not sure how may the results of the study translate into clinical practice, can something be said about that? The susceptibility testing was based on one culture limits the possible translation of the results into clinical practice.
Response: Thank you for this remark. This study gives information about a cell free supernatant of a probiotic co-culture. The supernatant itself had beneficial impact on the fecal murine microbiome without containing alive bacteria. Consequently, it may be useful in certain patient groups at risk for adverse events in case of probiotic therapy. Future animal models will have to assess the effect of CFS derived from OMNi BiOTiC® AAD10 in disease models. The Susceptibility testing was carried out in 5 repetitions, the presented data are a mean of these 5 approaches. However, they are based on 1 culture. This is a possible limitation and has been discussed as such in the limitations section of the discussion section.
Round 2
Reviewer 1 Report
The authors have extensively revised the manuscript. It is better and now any reader will be better educated on postbiotics, probiotics, and prebiotics. The statement that the present identifies, in vitro produced, potential postbiotics , is fairly clearly stated. The authors have pointed out and should realize that while in vitro experimentation can compliment in vivo work with bacteria, bacterial behavior and production of metabolites and structures is very different in vitro from in vivo and one must have very conservative evaluation of the impact of this approach. Doing animal model as well as clinical experimentation including straightforward simple analyses with good measurement metrics and data for evaluation, will provide the best information.
An important item of why these bacteria were chosen for this probiotic mixture is still incompletely presented and in NO reference published is this clarified. This reviewer as the authors wants well designed experimentation and measurements as this is a function of good quality journals and peer review and citation of these in PubMed. However, early reports of nutraceuticals is unfortunately in the non-scientific literature and these may only be idenrified by such as a Goggle search, these publications are NOT on PubMed. Even setting my search parameters even broader the available literature on this new Omni probiotic is lacking. This is of concern, I would suspect that at least safety evaluations have been done, not effectiveness, but I cannot find anything. WHY mix 10 bacteria, what is the purpose? Line 59 refers the reader to knowledge that these 10 bacterial types have all been identified to possess beneficial activities. There are hundreds more bacterial types that have probiotic activity but they are used individually or sometimes in combinations, as in VSL#3 there are investigated reasons why to combine bacterial types. It should at least be stated what reasons one might want to combined bacterial types to make a mixture with enhanced, greater, or different beneficial activities.
With the concept of postbiotic one might view these studies as identification of lead compounds with beneficial activity. This may be the case, it is not clearly stated but is implied, but then one again would wonder why this probiotic mixture has so many bacteria?
Knowledge about products and food items one consumes are important. For scientific evaluation, the reasons (1) behind all these many bacterial types and (2) need to combination, both MUST be included in the Introduction of this is an incomplete concept manuscript. The authors must also determine if ANY published data on safety is available in a form which is scientifically sound, even if from the company. A neutraceutical without any prior publication must present some data on this. This is interesting and important, the data is limited but of good quality and some data predicts some good beneficially effect. But this Omni mixture is so new,, this appears to be the first manuscript submitted to a scientific journal with peer review, so some greater information must be included on conceptualization.
Author Response
The authors have extensively revised the manuscript. It is better and now any reader will be better educated on postbiotics, probiotics, and prebiotics. The statement that the present identifies, in vitro produced, potential postbiotics , is fairly clearly stated. The authors have pointed out and should realize that while in vitro experimentation can compliment in vivo work with bacteria, bacterial behavior and production of metabolites and structures is very different in vitro from in vivo and one must have very conservative evaluation of the impact of this approach. Doing animal model as well as clinical experimentation including straightforward simple analyses with good measurement metrics and data for evaluation, will provide the best information.
Response: Thank you very much for this comment and the suggestions you made
An important item of why these bacteria were chosen for this probiotic mixture is still incompletely presented and in NO reference published is this clarified. This reviewer as the authors wants well designed experimentation and measurements as this is a function of good quality journals and peer review and citation of these in PubMed. However, early reports of nutraceuticals is unfortunately in the non-scientific literature and these may only be idenrified by such as a Goggle search, these publications are NOT on PubMed. Even setting my search parameters even broader the available literature on this new Omni probiotic is lacking. This is of concern, I would suspect that at least safety evaluations have been done, not effectiveness, but I cannot find anything. WHY mix 10 bacteria, what is the purpose? Line 59 refers the reader to knowledge that these 10 bacterial types have all been identified to possess beneficial activities. There are hundreds more bacterial types that have probiotic activity but they are used individually or sometimes in combinations, as in VSL#3 there are investigated reasons why to combine bacterial types. It should at least be stated what reasons one might want to combined bacterial types to make a mixture with enhanced, greater, or different beneficial activities.
With the concept of postbiotic one might view these studies as identification of lead compounds with beneficial activity. This may be the case, it is not clearly stated but is implied, but then one again would wonder why this probiotic mixture has so many bacteria?
Knowledge about products and food items one consumes are important. For scientific evaluation, the reasons (1) behind all these many bacterial types and (2) need to combination, both MUST be included in the Introduction of this is an incomplete concept manuscript. The authors must also determine if ANY published data on safety is available in a form which is scientifically sound, even if from the company. A neutraceutical without any prior publication must present some data on this. This is interesting and important, the data is limited but of good quality and some data predicts some good beneficially effect. But this Omni mixture is so new,, this appears to be the first manuscript submitted to a scientific journal with peer review, so some greater information must be included on conceptualization.
Response: We agree with the reviewer, that more information about the probiotic used would be beneficial. A study by Timmermann et al could clearly demonstrate a benefit of a multi-species (VSL#3 – which you also mentioned in this and your previous comments) probiotic over single strain or multi strain in the prevention of antibiotic associated diarrhea. As such a combination of synergistic probiotics seems beneficial.
As to previous applications we allow to refer to 6 publications dealing with the benefits of OMNi BiOTiC® AAD10. Of these Lang et al 2010 (google), Koning et al 2008 (google, PUBMED: PMID 17900321), van Wietmarschen et al 2020 (PUBMED: PMID 32404062) revealed reduced incidences of antibiotic associated diarrhea under treatment of OMNi BiOTiC® AAD10. Stadlbauer et al (PMID 30694100) found beneficial effects of OMNi BiOTiC® AAD10 if administered in the early phase of sepsis in 2019. Rabl et al could demonstrate a reduction of antibiotic associated diarrhea and diarrhea in oncological patients under treatment with OMNi BiOTiC® AAD10 (article in German). Zollner-Schwetz et al (PUBMED: PMID 32481668) showed a reduced colonization of the gut with multi-resistant bacteria in patients housed in long-term care facilities in 2020. As such AAD10 is not new at all. Different name-giving between the manufacturer (Winclove) and the distributor (Institut Allergosan) however make it difficult to find these articles in the literature. We should have given more information about previous applications in clinical studies in the manuscript.
OMNi BiOTiC® AAD10 (distributed by Institut AllergoSan (Graz, Austria) and produced by Winclove Probiotics B.V. (Amsterdam, The Netherlands) a GMP facility for manufacturing dietary supplements complying with NSF/ANSI standard 173-2008 and certified according to ISO 22000:2005).
We have included all this information in the introduction section of the second revision of the manuscript now.